# Association of School Instructional Mode with Community COVID-19 Incidence during August–December 2020 in Cuyahoga County, Ohio

**DOI:** 10.3390/ijerph21050569

**Published:** 2024-04-29

**Authors:** Pauline D. Terebuh, Jeffrey M. Albert, Jacqueline W. Curtis, Kurt C. Stange, Suzanne Hrusch, Kevin Brennan, Jill E. Miracle, Wail Yar, Prakash R. Ganesh, Heidi L. Gullett, Johnie Rose

**Affiliations:** 1University Hospitals Cleveland Medical Center, Cleveland, OH 44106, USA; 2School of Medicine, Case Western Reserve University, Cleveland, OH 44106, USA; jeffrey.albert@case.edu (J.M.A.); jacqueline.curtis@case.edu (J.W.C.); heidi.gullett@case.edu (H.L.G.);; 3Neighborhood Family Practice, Cleveland, OH 44102, USA; 4Cuyahoga County Board of Health, Parma, OH 44130, USA

**Keywords:** COVID-19, distance education, return to school, social determinants of health, community transmission

## Abstract

Remote and hybrid modes of instruction were employed as alternatives to in-person instruction as part of early mitigation efforts in response to the COVID-19 pandemic. We investigated the impact of a public school district’s instructional mode on cumulative incidence and transmission in the surrounding community by employing a generalized estimating equations approach to estimate the association with weekly COVID-19 case counts by zip code in Cuyahoga County, Ohio, from August to December 2020. Remote instruction only (RI) was employed by 7 of 20 school districts; 13 used some non-remote instruction (NRI) (2–15 weeks). Weekly incidence increased in all zip codes from August to peak in late fall before declining. The zip code cumulative incidence within NRI school districts was higher than in those offering only RI (risk ratio = 1.12, *p* = 0.01; risk difference = 519 per 100,000, 95% confidence interval (123–519)). The mean effect for NRI on emergent cases 2 weeks after mode exposure, controlling for Social Vulnerability Index (SVI), was significant only for high SVI zip codes 1.30, *p* < 0.001. NRI may be associated with increased community COVID-19 incidence, particularly in communities with high SVI. Vulnerable communities may need more resources to open schools safely.

## 1. Introduction

On 14 March 2020, a State of Emergency was declared in Ohio in response to the emerging COVID-19 pandemic [1]. A state executive order to prohibit mass gatherings was followed on 16 March 2020 by the closure of all Ohio K-12 schools and was quickly succeeded by a stay-at-home order and the closure of non-essential businesses [2]. Schools concluded the academic year in a remote instructional mode. During the spring, a cascade of mitigation policies and procedures was implemented in healthcare, congregate settings, and across the community in an effort to “flatten the curve”.

During the summer recess, Ohio K-12 public school districts, as well as private and parochial schools, utilized the experience accrued during the previous academic year to develop strategies and plans for delivering instruction during the upcoming 2020–2021 academic year. Strategies ranged from continuing fully remote instruction (RI) to 5-day in-person school instruction for all ages with mandated face coverings and social distancing procedures. Many schools developed options for students, such as rotating schedules or differential plans by grade level. Reporting staff and student COVID-19 cases was mandated by the state. Due to rising cases during the midsummer months and limited availability of COVID-19 testing for school-aged children, Cuyahoga County health officials recommended that school districts adopt a remote instructional mode for the start of the school year [3]. While private and parochial schools throughout the county draw students from varied geographic regions, the 31 public school districts in the county serve students within defined geographic boundaries. From August 2020 to December 2020, school districts adapted their instructional modes based on multiple considerations, including public health recommendations, the detection of cases within the school community or broader community transmission trends, staffing considerations, and the feasibility of implementing prevention measures. Concurrently, state and local recommendations for community mitigation also varied over the time period.

While the choice of instructional mode was multifactorial and complex, one consideration was the contribution of school instructional mode to overall community transmission trends. School closures were widely implemented as part of non-pharmaceutical pandemic response efforts during the spring of 2020. A number of studies attempted to estimate the relative contribution of school closures on COVID-19 incidence [4,5,6,7]. However, studies on the impact of school reopening largely focused on in-school transmission trends and the impact on staff and students [8,9,10]. A large national household survey found much greater odds of COVID-19-like illness among household members of persons attending in-person school, an effect that was attenuated when more mitigation measures were implemented within the attended school [11]. Most attempts to understand the impact of lifting restrictions and reopening schools on the broader community have relied on predictive modeling studies without comparison between communities undertaking differential approaches to the reopening of schools [12,13,14]. However, some studies have examined the association between instructional mode and COVID-19 outcomes. These studies, however, analyze COVID-19 outcomes at the county level while that county may be made up of heterogeneous school districts employing different instructional modes. One study found no association between instructional mode and COVID-19 hospitalizations when baseline circulation was low, but at higher levels of COVID-19 community circulation, the results were inconclusive [15]. We compare the incidence of COVID-19 in the community within geographic boundaries of the county’s school districts that employed differing instructional modes during the fall of the academic year of 2020–2021 in order to assess the impact of public-school instructional mode on the community incidence of COVID-19 in the surrounding community.

## 2. Materials and Methods

The Cuyahoga County Board of Health compiled and reported weekly (with data shared from the Cleveland Department of Health jurisdiction) COVID-19 case counts for each zip code in the county. Cases reported from August to December 2020 were used to calculate weekly population-adjusted incidence for residents of each zip code. The reporting period began each Wednesday from August to October and shifted to beginning each Sunday from November to December to align with the CDC Epi week. This frameshift resulted in a single shortened reporting period from Wednesday to Saturday to accommodate the frameshift at the end of October. Some zip code areas straddle public school district boundaries. Zip codes with <90% of its population lying within a single school district were excluded from the analysis. The aggregate zip code COVID-19 case counts utilized for this study were obtained from a public data dashboard made available on the Cuyahoga County Board of Health website, and therefore, this study was not considered human subjects research and did not require institutional review board approval.

Using U.S. Census Tract level data from the Center for Disease Control and Prevention’s Social Vulnerability Index (SVI) (Centers for Disease Control and Prevention/Agency for Toxic Substances and Disease Registry/Geospatial Research, Analysis, and Services Program; CDC Social Vulnerability Index 2018), a population-weighted mean SVI was calculated for each zip code. The SVI uses 15 U.S. census variables to assign an overall vulnerability rank of a population to public health emergencies based on factors such as socioeconomic status, household composition and crowding, disability, minority status, and access to transportation and ranks the population on a scale of 0 (least vulnerable) to 1 (most vulnerable) [16,17,18]. See Appendix A for SVI details. For the U.S. Census Tracts that straddle zip code lines, the tract population was assigned to the zip code in which the majority of the population resides. In cases in which the tract population was divided evenly between zip codes (closer than 60:40), half the population was assigned to each zip code.

The public school district instructional mode for each week was characterized as defined by the Ohio Department of Education [19]:Five-day in-person: all students have the option of in-person instruction, even if schedules are somewhat adjusted;Fully remote: all students receive only remote education, which may be teacher-led instruction or student-led paced learning;Hybrid: a mix of in-person and remote education, noting some grade levels may be entirely in-person or entirely remote.

The public school district instructional mode was ascertained by calendars and documents published on the school district website. No distinction was made between closure for holidays or administrative days and remote instruction. For weeks during which closure or RI occurred on some days while hybrid or 5-day in-person instruction occurred on others (i.e., the week encompassing Labor Day and the week encompassing Thanksgiving), the instructional mode was designated as either 5-day in-person or hybrid instruction. For some analyses, instructional mode was dichotomized as either RI or non-remote instruction (NRI).

### Statistical Analysis

The analysis was performed using SAS software 9.4 (University Edition, SAS Institute, Cary, NC, USA). Given the heterogeneity of the hybrid instructional mode, the hybrid instruction category was combined with the 5-day in-person category mode to dichotomize exposure to RI and NRI. Population-adjusted cumulative incidence of each included zip code population was used to calculate a relative risk and risk difference for districts that employed some NRI during the study period in comparison to districts that maintained RI during the entire study period. Cumulative incidence for the pre-study period of March–July 2020 was used as a reference period during which no NRI took place in any school district. A *t*-test at a significance level of 0.05 was calculated using openepi.com. 

We employed a generalized estimating equations (GEE) approach to estimate the association between the instructional mode of the public school district and emergent COVID-19 cases in the community at the zip code level over the 22-week observation period. The outcome was population-adjusted weekly zip code COVID-19 case incidence. Because of the lag time between an exposure that could lead to transmission of the virus and the detection and reporting of a resulting COVID-19 case, the instructional mode from 2 weeks prior (2-week lag) was defined as a lagged covariate. We used GEE to fit a log-linear model, assuming a Poisson-distributed outcome (COVID-19 incidence), with the (log-transformed) zip code population as an offset, and using a first-order autoregressive working correlation structure. The covariates included the (lagged) instructional mode, week (as a categorical variable), and SVI, dichotomized to low (≤0.5) or high (>0.5). A mode by SVI interaction term was added in a second model. GEE regression parameter estimates were obtained with robust estimates of standard errors, and score tests were conducted. From exponentiated parameter estimates, we obtained estimated relative (population-adjusted) incidence rates (for example, of NRI versus RI modes) and corresponding 95% confidence intervals. An advantage of GEE is that it models the marginal—that is, the population or subpopulation, as opposed to individual (including individual district) level—effects, and thus is relevant to our public health question. Also, GEE is robust to departures from assumed distributions and avoids extraneous model assumptions of other (e.g., mixed model) approaches [20].

A sensitivity analysis was conducted, including all county zip codes in the model. These additional zip codes (zip codes excluded from the main analysis) crossed school district boundaries, with a portion of their zip code population residing in more than one school district, and were included in the sensitivity analysis in association with each school district’s instructional mode to which a portion of the zip code population was exposed.

## 3. Results

Of the 51 zip codes within Cuyahoga County, 37 are contained largely within a single school district; the remaining 14 zip codes straddle more than one school district and were excluded. Of the 31 school districts, 20 have at least one zip code contained within its boundaries not shared with a neighboring school district; 11 school districts did not have a zip code contained within its boundaries not also shared (≥10%) by a neighboring district and were therefore not reflected in the analysis (Figure 1, Table 1). Population-adjusted weekly incidence trends increased in all zip codes from August to peak in late November to early December before declining in late December (Figure 2). Weekly incidence ranged from 0 to 1164 per 100,000.

Seven school districts remained in RI mode during the entire study period; ten school districts employed RI and some hybrid instruction; three school districts employed periods of RI, hybrid, and 5-day in-person instruction mode (Figure 2). Of the school districts that employed some NRI, the periods of NRI mode varied among school districts (hybrid (2–15 weeks); 5-day in-person (4–6 weeks)). 

The population-weighted zip code SVI ranged from 0.02 to 0.95 (Appendix A). For included school districts that attempted some NRI, average zip code SVI (0.33; range = 0.02–0.70) was significantly lower than zip codes in districts that remained in RI (0.73; range = 0.27–0.95) for both the pre-study and study periods (*p* < 0.001).

The mean population-adjusted cumulative incidence during the study period for zip codes within school districts that employed some NRI was 4752 per 100,000, while the cumulative incidence for zip codes within school districts that employed only RI was 4243 per 100,000, resulting in a risk ratio of 1.12, *p* = 0.01. There was a significant average zip code risk difference of 509 per 100,000 with a 95% confidence interval (123–519) for schools that employed some NRI (Figure 3a). During the pre-study period (March–July 2020), in contrast, zip codes in districts that employed some NRI during the study period had a lower average cumulative incidence (918 per 100,000) than the school districts that would remain in RI (1295 per 100,000) (Figure 3b) while showing a much greater increase in incidence when comparing within zip code incidence during the study period to its incidence during the pre-study period (incident rate ratio = 0.71 for pre-study cumulative incidence in NRI zip codes: pre-study cumulative incidence in RI zip codes).

The mean weekly zip code incidence for RI versus NRI mode (as characterized by the mode during the week beginning two weeks prior) is shown in Figure 4. The GEE parameter estimates for the model showed a mean effect of NRI mode (estimated weekly incidence rate ratio for NRI versus RI mode) of 1.12 (*p* = 0.03). With the introduction of SVI into the model, which shows a significant association of SVI with incidence, the result is a non-significant association between instructional mode and zip code incidence overall. However, there is a significant instructional mode by SVI interaction (*p* = 0.006). When dichotomizing SVI, the effect of instructional mode for zip codes with low SVI (≤0.5) is not significant in the model; however, for high SVI (>0.5), the effect of mode is statistically significant (*p* < 0.001) with an effect (incidence rate ratio) estimate of 1.3 for NRI versus RI. The sensitivity analysis, including all 51 zip codes and all 31 school districts, showed a similar trend, although the effect estimate was smaller (1.2) for NRI (*p* = 0.03). The results are summarized in Table 1.

## 4. Discussion

Safely reopening schools has become a major priority for public health and educational decision makers and for society in general. While reporting of student and staff school-associated COVID-19 cases has been a foundational metric for monitoring safety [8,9,21,22], the potential contribution of NRI to community transmission has also been a concern but has been difficult to assess. The heterogeneous approach to the instructional mode during the Fall 2020 academic period in Cuyahoga County allowed for a comparison of community incidence by instructional mode. While COVID-19 incidence increased in all communities to peak between late November and December, the increases were greater for communities that employed NRI, i.e., some periods of hybrid or full in-person instruction. Populations living in a public school district that employed some non-remote instruction overall had 12% more cases of COVID-19 during the fall of 2020. When modeling the effect of the instructional mode on emerging cases after a 2-week lag to account for an incubation period, testing, and reporting delay, NRI was associated with an 11% higher incidence rate during that reporting week in comparison to populations living in school districts that two weeks prior had delivered only remote instruction. However, after controlling for SVI in this model, only school districts with higher social vulnerability that attempted NRI had an associated increase in community incidence of COVID-19.

The communities represented in the analysis had a very wide range of social vulnerability, as represented by SVI, a measure intended to reflect socioeconomic status, household composition, minority status and language, housing type, and transportation access. Surveillance data and studies have shown that high SVI has been associated with higher incidence and greater morbidity [23], and this association was found in Cuyahoga County during the March–July 2020 period. Low SVI communities had lower COVID-19 incidence during the early part of the pandemic, were more likely to attempt NRI, and were more likely to have a higher incidence of COVID-19 during the fall. The inverse association was found for high SVI communities. They had a relatively higher incidence of COVID-19 (in comparison to low SVI zip codes) during the early pandemic, were more likely to remain in RI mode throughout the fall, and had a relatively lower incidence during the fall.

Our study had a number of limitations; in particular, the determination of weekly incidence is dependent on case ascertainment. Many factors, such as availability, access, and motivation for testing, can affect case counts. Whether attendance in school may directly contribute to increased transmission and, in turn, amplify community incidence or whether school attendance affects indirect factors via their effect on the family work environment or out-of-school interactions cannot be assessed in the current study. We used a 2-week lag to assess the differential impact of instructional mode on COVID-19 incidence as part of our model; however, the true lag may be different than two weeks. We coupled this model with a comparison of cumulative incidence over the entire study period, and both approaches yielded similar estimates. A mechanistic modeling approach, for example, focusing on the reproduction number, as in Nash et al. [24], might be considered as an alternative, possibly more dynamic, way of addressing our study question. The differential impact of school closure on community-level mobility and the average number of daily contacts has been explored in studies [25,26], but school reopening is likely to occur in tandem with other social mixing practices within a community, and an attempt to assess its impact out of context may not be meaningful. A study across Texas found a county-level association of in-person instruction with COVID-19 cases and deaths, as well as a large increase in adult mobility associated with in-person schooling. The temporal association suggested a potential “spillover” effect of school reopening due to increased mobility and changes in perception of which activities were deemed safe [27]. The option of remote work is likely greater on average for higher-earning parents who are residents of wealthier communities than for parents in lower-wage jobs who reside in poorer communities. At the same time, the resumption of in-person schooling may signal that it is safe to resume extracurricular activities or social gatherings. A study of individual risk factors for COVID-19 among children and adolescents in Mississippi found an association between having a positive COVID-19 test and recent attendance at a social gathering with persons outside the household, but not with in-person attendance at school or childcare [28]. 

Communities whose school districts chose to offer NRI may have differed in their attitudes and practices of masking and social distancing, SARS-CoV-2 test seeking or in other unmeasured ways that contributed to increased incidence during the fall of 2020. A number of studies have found that differences in adherence to mitigation practices were often associated with factors such as partisanship [29,30]. The effect of these differential attitudes may have grown over time with COVID-19 incidence increasing outside of institutional settings. While cases in congregate living settings contributed substantially to case counts during the early months of the pandemic, reports of community-associated cases became more common during the fall of 2020, including family gatherings during the holidays. These changes in transmission patterns were reflected in county surveillance data that showed a growing proportion of cases among younger age groups [31].

Since COVID-19 cases are reported by zip code, and their geographic area does not necessarily align with public school district boundaries, the population residing in excluded zip codes was not represented in the analysis. Likewise, some school districts were not represented because all of their zip codes overlapped with neighboring districts. However, the inclusion of all county zip code populations in the sensitivity analysis showed similar trends in the effect of instructional mode and SVI. These added zip codes had portions of their population in more than one school district, so the effect of multiple instructional modes combined into a single weekly zip code incidence would be expected to dilute the effect. Furthermore, teachers and staff may live in a different community from where they work. Additionally, private school instructional mode was not considered as the catchment area for their students is not geographically defined. 

While the relative contribution of the direct and indirect effects of NRI and the other factors associated with communities that attempted NRI during the fall of 2020 cannot be assessed, a temporal association between NRI and subsequent increases in COVID-19 cases in communities with fewer resources was observed. In addition, communities that attempted NRI tallied more COVID-19 cases overall during the fall than communities that adhered to remote instruction. Those same communities had relatively fewer cases before the beginning of the academic year and were, on average, the more resourced communities. While the finding that wealthier communities had a higher incidence of COVID-19 during the fall of 2020 was surprising, it corresponded with a phase of the pandemic during which cases and outbreaks began emerging more commonly from the community at large as opposed to within congregate and other high-risk settings, albeit still at relatively low incidence overall. Having weathered many months of the pandemic lockdowns, attempts at NRI may have been an indicator that communities were willing to accept greater risk, especially in more resourced communities, even though a close temporal association (a 2-week lag) between instructional mode and a surge in community cases could not be established. 

## 5. Conclusions

The detection of temporal coupling of instructional mode and community incidence in lower-resourced communities reinforces the need for an equity-grounded approach that provides greater support in order to be able to open safely. During 2021, the period after this study, access to vaccines, vaccine hesitancy, and the resultant vaccine coverage were newly emergent factors that likely affected community transmission trends as schools endeavored to resume NRI. Since vaccine eligibility for children under age 12 lagged behind that of older children, the potential contribution of school instructional mode to community transmission and the differential impact of social vulnerability and vaccine coverage was highly salient as new SARS-CoV-2 variants emerged. When evaluating non-pharmaceutical interventions for the control of emerging diseases, the social vulnerability and geographic characteristics of the affected community must be considered.

## Figures and Tables

**Figure 1 ijerph-21-00569-f001:**
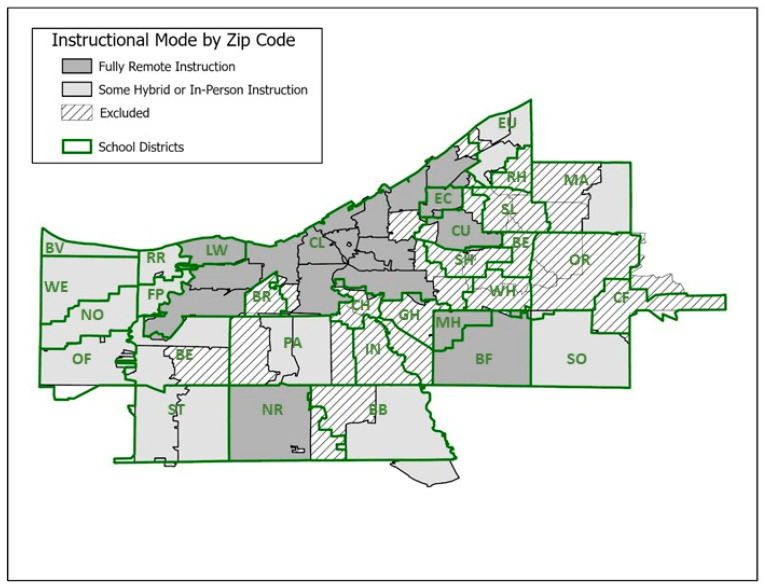
Map of county public school districts and zip codes. Zip codes straddling school district boundaries were excluded from the analysis. Zip codes in school districts with some hybrid or in-person instruction during the Fall of 2020 are shown in light grey. Zip codes in school districts with fully remote instruction during this time period are shown in dark grey.

**Figure 2 ijerph-21-00569-f002:**
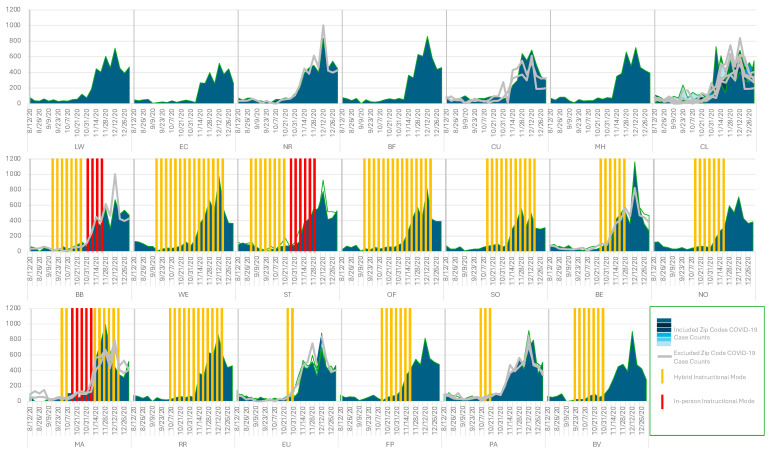
Weekly incidence of COVID-19 (per 100,000) during the fall of 2020 by public school district and zip code. Zip codes that straddle more than one school district were excluded from the analysis and shown with grey lines. Zip codes contained within a single school district are shown in shades of blue. The top row depicts zip codes that had remote instruction only. The vertical bars in the bottom two rows represent the school district’s weekly instructional mode (in-person or hybrid) during the observation period.

**Figure 3 ijerph-21-00569-f003:**
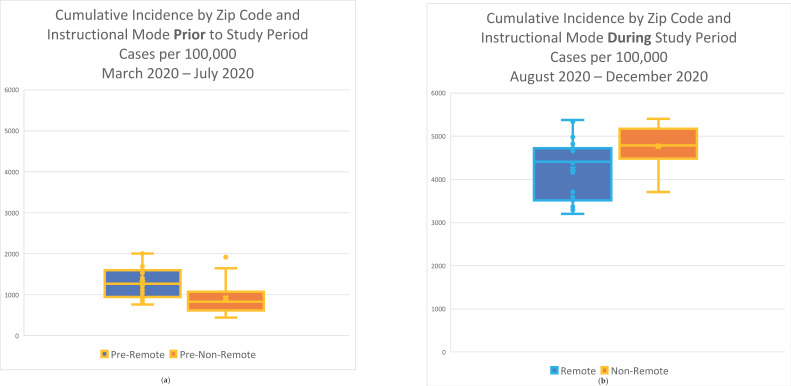
(**a**) The cumulative incidence of COVID-19 in zip codes employing some non-remote instruction during the study period was higher than in zip codes remaining in remote instructional mode during the same period; (**b**) the cumulative incidence of COVID-19 in zip codes that would employ non-remote instruction during the study period was lower during the pre-study period than in zip codes that would remain in remote instructional mode during both pre-study and study periods.

**Figure 4 ijerph-21-00569-f004:**
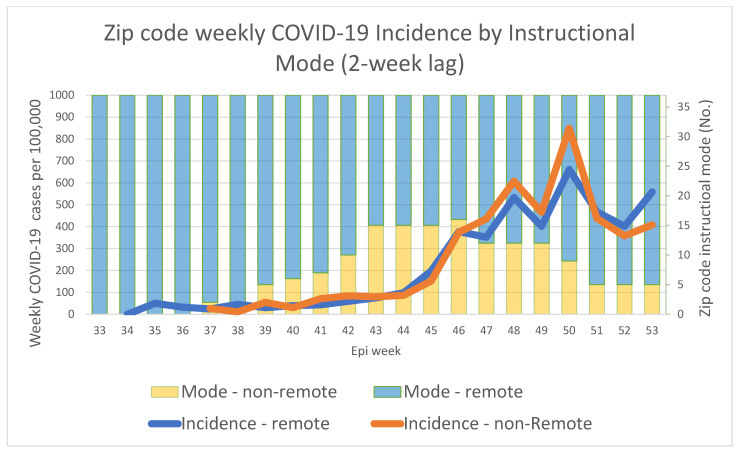
Zip code instructional mode is characterized by the mode during the school week beginning two weeks prior to the beginning of the COVID-19 case reporting week to account for a 2-week lag between instructional mode exposure and the reporting of incident cases. Average weekly population-adjusted zip code COVID-19 case incidence is shown by mode. Note: The first day of the case reporting week was Wednesday from August to October and Sunday from November to December.

**Table 1 ijerph-21-00569-t001:** Summary of analysis.

Unit of Analysis		
Exposure = Instructional Mode		
School districts (N = 31) ^1^		
Included	20	
NRI	13	
RI	7	
Excluded	11	
Outcome = COVID-19 Case Counts		
Zip codes (N = 51) ^2^		
Included	37	
Excluded	14	
Population Characteristics		
SVI (Range, 0.02–0.95) ^3^		
NRI	0.33 (0.01–0.70)	*p* < 0.001
RI	0.73 (0.27–0.95)	
Outcome		
Pre-study period Cumulative Incidence (per 100,000), NRI vs. RI		
Pre-NRI	918	RR = 0.71, *p* = 0.01 RD = −377, 95%CI ((−644)–(−112))
Pre-RI	1295	
Study period Cumulative Incidence (per 100,000), NRI vs. RI		
NRI	4752	RR = 1.12, *p* = 0.01 RD = 509, 95%CI (123–519)
RI	4243	
GEE model of association of Instructional Mode and COVID-19 Incidence, NRI vs. RI		
Without SVI interaction term		RR = 1.12, *p* = 0.03
With SVI > 0.5		RR = 1.3, *p* < 0.001
With SVI ≤ 0.5		RR = 1.0

^1^ School districts were excluded if all zip codes within the district were shared with neighboring districts. Included school districts that remained in remote instructional mode during the entire study period were categorized as RI; included school districts with any non-remote instructional mode were categorized as NRI. ^2^ Zip codes straddling more than one school district (>10%) were excluded from the analysis. ^3^ Social Vulnerability Index estimate was population weighted by U.S. Census Tract within each zip code. U.S. Census Tract populations that straddled zip codes were assigned to the majority zip code unless the population majority was evenly divided (within 60:40). Those census populations were split between the two zip codes. RI = remote instruction; NRI = non-remote instruction; SVI = Social Vulnerability Index; RR = risk ratio, rate ratio; RD = risk difference CI = confidence interval; GEE = general estimating equations.

## Data Availability

COVID-19 case counts were derived from public health reporting posted on the Cuyahoga County Board of Health’s COVID-19 dashboard. https://ccbh.net/covid-19-dashboard-and-resources/. U.S. Census Tract level data from the Center for Disease Control and Prevention’s Social Vulnerability Index (SVI) (Centers for Disease Control and Prevention/Agency for Toxic Substances and Disease Registry/Geospatial Research, Analysis, and Services Program. CDC Social Vulnerability Index 2018), https://www.atsdr.cdc.gov/placeandhealth/svi/index.html (accessed on 9 March 2021).

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
