# Peer review of "Association of School Instructional Mode with Community COVID-19 Incidence during August–December 2020 in Cuyahoga County, Ohio"

_ijerph, 2024, doi:10.3390/ijerph21050569_

Round 1
Reviewer 1 Report
Comments and Suggestions for Authors
This paper presents a detailed description of School Instruction Mode with Community COVID-19 During August to December 2020 in Cayahoga County, Ohio. For the most part the authors did an adequate job of reporting what they did and the results of the study. I have a few comments. First, it seems that the SVI is an important factor in the analysis of the data. You state that it uses 15 census variables to assign an overall rank of a population. I would like to know what those 15 variables are. I also would be interested in knowing the range of value of the different variables on the impact on COVID-19 or not. I also would like to know COVID-19 rates over those zip codes. I think you have data for a much more interesting focus of your study. I think your results match rather well to what one would predict, but what are the driving factors making up the SVI.
The colors you chose for Figure 1 make it difficult to identify the regions. The boundary colors for both school districts and zip codes appear to be the same color. Figure 2 is unreadable. Both figures need improvement. The axis labels and legends in Figure 4 are difficult to read.
Were there other factors, such as weather that had an impact on your results? Your result section on page 9 was difficult to read. A table would make the results clearer. I had to make a table to understand the results. make it clear what the components of your results were. There are many variables are involved in your different results. I would think population, incidence of COVID, vaccination, NRI and RI, and hybrid are all factors. Can you create a table that shows these results?
I would like to see recommendations be stated clearly as part of the conclusions. I think paper could be made better with these changes.
Author Response
This paper presents a detailed description of School Instruction Mode with Community COVID-19 During August to December 2020 in Cayahoga County, Ohio. For the most part the authors did an adequate job of reporting what they did and the results of the study. I have a few comments. First, it seems that the SVI is an important factor in the analysis of the data. You state that it uses 15 census variables to assign an overall rank of a population. I would like to know what those 15 variables are. I also would be interested in knowing the range of value of the different variables on the impact on COVID-19 or not. I also would like to know COVID-19 rates over those zip codes. I think you have data for a much more interesting focus of your study. I think your results match rather well to what one would predict, but what are the driving factors making up the SVI.
Thank you for your careful review. As requested, all 15 variables and additional documentation on Social Vulnerability Index (SVI) has been added to a Supplement. The SVI has been developed as a public health planning and preparedness tool for the distribution of resources. “Social vulnerability refers to the potential negative effects on communities caused by external stresses on human health.” Disease outbreaks were one of the scenarios for which the SVI was created. The composite SVI reflects a census tract’s percentile rank ranging from 0 to 1, with a higher value corresponding to greater vulnerability. The composite SVI is constructed by combining the rankings of the 15 individual U.S. census variables listed below.
.
- Socioeconomic Status
- Below Poverty
- Unemployed
- Income
- No High School Diploma
- Household Characteristics
- Aged 65 & Older
- Aged 17 & Younger
- Civilian with a Disability
- Single-Parent Households
- English Language Proficiency
- Racial & Ethnic Minority Status
- Hispanic or Latino (of any race); Black and African American, Not Hispanic or Latino; American Indian and Alaska Native, Not Hispanic or Latino; Asian, Not Hispanic or Latino; Native Hawaiian and Other Pacific Islander, Not Hispanic or Latino; Two or More Races, Not Hispanic or Latino; Other Races, Not Hispanic or Latino
- Housing Type & Transportation
- Multi-Unit Structures
- Mobile Homes
- Crowding
- No Vehicle
- Group Quarters
An analysis of each of these variables individually may have been possible, but the SVI was developed because of the interaction of these variables contributes to a community’s overall vulnerability. We found that communities with attempts at in-person schooling in communities with higher SVI were associated with increased COVID-19 incidence whereas that association was not found in low SVI communities. That finding has implications for resource allocation and the choice of non-pharmaceutical interventions.
- CDC/ATSDR Social Vulnerability Index. https://www.atsdr.cdc.gov/placeandhealth/svi/index.html
- CDC/ATSDR SVI Data and Documentation Download. https://www.atsdr.cdc.gov/placeandhealth/svi/data_documentation_download.html
- Flanagan B, Gregory E, Hallisey E, et al. A social vulnerability index for disaster management. Journal of Homeland Security and Emergency Management. 2011;8(1). https://www.atsdr.cdc.gov/placeandhealth/svi/img/pdf/Flanagan_2011_SVIforDisasterManagement-508.pdf
The colors you chose for Figure 1 make it difficult to identify the regions. The boundary colors for both school districts and zip codes appear to be the same color. Figure 2 is unreadable. Both figures need improvement. The axis labels and legends in Figure 4 are difficult to read.
We have updated all of the figures and hope that helps with clarity.
Were there other factors, such as weather that had an impact on your results? Your result section on page 9 was difficult to read. A table would make the results clearer. I had to make a table to understand the results. make it clear what the components of your results were. There are many variables are involved in your different results. I would think population, incidence of COVID, vaccination, NRI and RI, and hybrid are all factors. Can you create a table that shows these results?
The table that presented SVI values has been moved to the Supplement replaced with the table that was requested summarizing the units of analysis, population characteristics, and salient outcomes. Assessing the impact of weather was beyond the scope of this analysis and would have been similar over the geographic area under study.
I would like to see recommendations be stated clearly as part of the conclusions. I think paper could be made better with these changes.
We have added additional clarification to the conclusions.
Thank you again for your thoughtful review. We hope these improvements and clarifications address your concerns.
Reviewer 2 Report
Comments and Suggestions for Authors
The study presents a compelling topic and demonstrates a strong methodological foundation. However, several minor issues were identified during the review process that warrant attention.
Figure 2 lacks clarity and fails to provide informative visuals. Enhancing both the scale and quality of the figure is necessary to ensure readability and comprehension.
While the authors assert that the utilization of open data negated the need for ethical review, it is imperative to note that obtaining permission for an exemption ethical review procedures from the local committee or commission is essential protocol. If such permission was not required, there should be documentation indicating a decision by the organization's leadership that exempted this study from ethical review in any form.
These comments primarily pertain to technical aspects of the manuscript.
Author Response
The study presents a compelling topic and demonstrates a strong methodological foundation. However, several minor issues were identified during the review process that warrant attention.
Figure 2 lacks clarity and fails to provide informative visuals. Enhancing both the scale and quality of the figure is necessary to ensure readability and comprehension.
We have updated all of the figures and hope that helps with clarity.
While the authors assert that the utilization of open data negated the need for ethical review, it is imperative to note that obtaining permission for an exemption ethical review procedures from the local committee or commission is essential protocol. If such permission was not required, there should be documentation indicating a decision by the organization's leadership that exempted this study from ethical review in any form.
We have added an IRB Statement for further clarifications. The Cuyahoga County Board of Health follows privacy guidelines when releasing data to the public. They require a minimum number of aggregate case counts per zip code prior to public release in order to protect privacy. The COVID-19 aggregate case counts far exceeded those thresholds.
Institutional Review Board Statement: Aggregate COVID-19 case counts by zip code were obtained from Cuyahoga County Board of Health COVID-19 dashboard on its public website and did not involve data collection from human subjects or access to any such data and was therefore not considered human subjects research.
We are following definitions in official NIH guidance below:
“An individual is considered to be conducting human subjects research when for research purposes, he or she, e.g.:
- Obtains consent from subjects;
- Interacts or intervenes with subjects; or
- Conducts activities with identifiable data or specimens. Note: Coded data or are considered identifiable, if the investigator or any member of the research team has access to the code key.”
Thank you again for your thoughtful review. We hope these improvements and clarifications address your concerns.
Reviewer 3 Report
Comments and Suggestions for Authors
Summary: Thank you for the opportunity to review the study by Terebuh et al. I have studied both the effectiveness of non-pharmaceutical interventions and infection control measures at the population- and individual level. I am thus receptive to studies undertaking similar investigations and I believe there is still much to be learnt, including the impact of school instruction mode on community incidence as presented in this study. Overall, I see merit in the analysis by Therebuh et al. but I have major concerns, especially regarding the statistical analysis, which I describe below. Before these concerns are addressed, I am not sure that any conclusions can be drawn from this study. To be clear, I am indifferent about the outcome of the revised analysis – I also see value in negative findings.
Abstract: The abstract requires revision. Important information is missing or not clearly presented. For example, the total number of analyzed districts is not clear. Uncertainty for the risk ratio is not reported. The main results should be reported in a clearly structured way, here between population comparison and pre/post comparison, which was difficult to figure out from the abstract in its current form. Not sure if the SVI needs to be reported if word limit is an issue; the association of SVI with the outcome suffices.
Introduction: In contrast to the Abstract, I commend the authors for an introduction that was very well written, clearly structured, and comprehensive!
Materials and Methods – Exclusion criteria: The exclusion criteria for the main analysis seem strict, although I acknowledge that the authors perform a sensitivity analysis with all zip codes included. Why was <90% chosen in the main analysis? Would a slightly lower cutoff include much more zip codes in the diffult analysis? Would more zip codes permit more distinction in the exposure groups (see comment further below)? More importantly though, I missed information on why there can be school districts not containing zip codes and why they are excluded. If they don’t contain a zip code, does that mean there is also no case reporting? Otherwise, it seems unreasonable to throw their data away just because they have no SVI.
Materials and Methods – Definitons: As far as I can see, the “NRI” abbreviation is introduced for the first time in the statistical analysis without prior mention of what it is (except for the Abstract). I would mention the definition of RI and NRI clearly at first occurrence in the manuscript (excluding Abstract) and make sure both appear at the same time to improve readability.
Materials and Methods – Statistical analysis: Here lies my greatest concern. First of all, the analysis can be divided into an exposed/unexposed population comparison (exploiting variation between populations: analysis A1) and a pre/post comparison (exploiting variation over time: analysis A2). It would be helpful to have descriptive figure showing for each school district the instructional mode over time (a colored tile plot or something), to get a feeling of what analysis has greater potential. Based on my reading, I guess there is more potential in A1, so let me start with this analysis.
Materials and Methods – Analysis A1: Comparing the cumulative incicende between the RI and the NRI group during the study period is perfectly fine. You could also use a Poisson model here and adjust for SVI and, importantly, pre-study cumulative incidence per school district. The latter can be used as an independent variable or you can directly compute the difference between the pre- and post-study incidence and use this as the outcome. Since the pre-study incidence in the summer was rather low (judging from Figure 3), I think this adjustment will not have much of an impact on the results. It may have greater impact on the narrative in the discussion, where I think it is anyway quite speculative to say that lower incidence in the summer period predicted less RI adoption in the fall period followed by higher incidence.
Materials and Methods – Analysis A2: The pre/post comparison within districts is more difficult and, depending on how much potential there actually is, you may prefer to leave it out. Either way, I am not convinced by the GEE model. In my experience, these models are often difficult to fit and not very robust. I think there are better, more common approaches to conduct the pre/post assessment. Common methodologies have been summarized in this literature review: https://link.springer.com/article/10.1007/s10654-022-00908-y. What might be most suitable to the authors is to first estimate each school district’s reproduction number Rt and then associate Rt with the presence of different instructional modes. Thereby, you can circumvent the fixed 2-week lag assumption, which otherwise demands a sensitivity analysis. Since you have weekly aggregated data, you may try the recently developed approach by Nash et al. in PLoS Comput Biol (https://journals.plos.org/ploscompbiol/article?id=10.1371/journal.pcbi.1011439), which extends the popular EpiEstim package to handle temporally aggregated data. Note also that there are other latent variable approaches that directly incorporate uncertainty in the estimation of Rt (see the review article above), but I admit that these approaches are more involved and maybe out of scope.
Materials and Methods – Continuous exposure: Since school districts followed RI or NRI to a varying extent, have you also tested continuous exposures, e.g. the number of weeks with RI or NRI? Could be an additional/sensitivity analysis for the first analysis A1. Also, if you include all school districts, does a comparison between in-person, fully remove, and hybrid become feasible?
Results – SVI: Is the SVI dichotomization with a 0.5 cutoff common or are there empirically more justified cutoffs? Do you even need a cutoff or could you not just estimate the interaction between the continuous SVI and the binary instruction mode variable?
Figures: Not sure what went wrong, but the quality of the figures is poor, especially for Figure 1 and 2, but also the others. I could only guess what was shown.
Table 1: Could be moved to the appendix.
I hope my comments were helpful and I wish the authors best of luck with any revisions.
Comments on the Quality of English Language
English is fine, and probably better than mine, considering native speakers among the authors.
Author Response
Summary: Thank you for the opportunity to review the study by Terebuh et al. I have studied both the effectiveness of non-pharmaceutical interventions and infection control measures at the population- and individual level. I am thus receptive to studies undertaking similar investigations and I believe there is still much to be learnt, including the impact of school instruction mode on community incidence as presented in this study. Overall, I see merit in the analysis by Therebuh et al. but I have major concerns, especially regarding the statistical analysis, which I describe below. Before these concerns are addressed, I am not sure that any conclusions can be drawn from this study. To be clear, I am indifferent about the outcome of the revised analysis – I also see value in negative findings.
Abstract: The abstract requires revision. Important information is missing or not clearly presented. For example, the total number of analyzed districts is not clear. Uncertainty for the risk ratio is not reported. The main results should be reported in a clearly structured way, here between population comparison and pre/post comparison, which was difficult to figure out from the abstract in its current form. Not sure if the SVI needs to be reported if word limit is an issue; the association of SVI with the outcome suffices.
We very much appreciate your thorough and thoughtful review.
We have reorganized the Abstract in response to the above suggestions.
Introduction: In contrast to the Abstract, I commend the authors for an introduction that was very well written, clearly structured, and comprehensive!
Materials and Methods – Exclusion criteria: The exclusion criteria for the main analysis seem strict, although I acknowledge that the authors perform a sensitivity analysis with all zip codes included. Why was <90% chosen in the main analysis? Would a slightly lower cutoff include much more zip codes in the diffult analysis? Would more zip codes permit more distinction in the exposure groups (see comment further below)? More importantly though, I missed information on why there can be school districts not containing zip codes and why they are excluded. If they don’t contain a zip code, does that mean there is also no case reporting? Otherwise, it seems unreasonable to throw their data away just because they have no SVI.
We acknowledge that the 90% threshold was chosen and other thresholds could have been used. We did perform a sensitivity analysis utilizing all zip codes without any threshold and found similar results. We have attempted to clarify the exclusion of school districts in the manuscript.
“Of the 31 school districts, 20 have at least one zip code contained within its boundaries not shared (≥10%) with a neighboring school district; 11 school districts did not have a zip code contained within its boundaries not also shared by a neighboring district and were therefore not reflected in the analysis. (Figure 1)”
Population-weighted SVI was calculated for every zip code. While some of these zip codes and school districts were included in the primary analysis, all zip codes and school districts were included in the sensitivity analysis.
Materials and Methods – Definitons: As far as I can see, the “NRI” abbreviation is introduced for the first time in the statistical analysis without prior mention of what it is (except for the Abstract). I would mention the definition of RI and NRI clearly at first occurrence in the manuscript (excluding Abstract) and make sure both appear at the same time to improve readability.
This has been corrected.
Materials and Methods – Statistical analysis: Here lies my greatest concern. First of all, the analysis can be divided into an exposed/unexposed population comparison (exploiting variation between populations: analysis A1) and a pre/post comparison (exploiting variation over time: analysis A2). It would be helpful to have descriptive figure showing for each school district the instructional mode over time (a colored tile plot or something), to get a feeling of what analysis has greater potential. Based on my reading, I guess there is more potential in A1, so let me start with this analysis.
We hope that the re-worked Figure 2 makes the instructional mode over the study period more clear.
Materials and Methods – Analysis A1: Comparing the cumulative incicende between the RI and the NRI group during the study period is perfectly fine. You could also use a Poisson model here and adjust for SVI and, importantly, pre-study cumulative incidence per school district. The latter can be used as an independent variable or you can directly compute the difference between the pre- and post-study incidence and use this as the outcome. Since the pre-study incidence in the summer was rather low (judging from Figure 3), I think this adjustment will not have much of an impact on the results. It may have greater impact on the narrative in the discussion, where I think it is anyway quite speculative to say that lower incidence in the summer period predicted less RI adoption in the fall period followed by higher incidence.
The school districts that chose non-remote instruction in the fall had a lower pre-study cumulative incidence than the school districts that remained in RI.
Materials and Methods – Analysis A2: The pre/post comparison within districts is more difficult and, depending on how much potential there actually is, you may prefer to leave it out. Either way, I am not convinced by the GEE model. In my experience, these models are often difficult to fit and not very robust. I think there are better, more common approaches to conduct the pre/post assessment. Common methodologies have been summarized in this literature review: https://link.springer.com/article/10.1007/s10654-022-00908-y. What might be most suitable to the authors is to first estimate each school district’s reproduction number Rt and then associate Rt with the presence of different instructional modes. Thereby, you can circumvent the fixed 2-week lag assumption, which otherwise demands a sensitivity analysis. Since you have weekly aggregated data, you may try the recently developed approach by Nash et al. in PLoS Comput Biol (https://journals.plos.org/ploscompbiol/article?id=10.1371/journal.pcbi.1011439), which extends the popular EpiEstim package to handle temporally aggregated data. Note also that there are other latent variable approaches that directly incorporate uncertainty in the estimation of Rt (see the review article above), but I admit that these approaches are more involved and maybe out of scope.
We appreciate the reviewer’s misgivings about our chosen method and suggestions for alternative approaches. We would respectively like to point out some advantages of the GEE approach, and have tried in our revision to better justify its use. First, GEE models the marginal – that is, the population or subpopulation, as opposed, to individual (including individual district) level – effects, and thus is relevant to our public health question. Second, it tends to be robust to departures from assumed distributions and avoids extraneous model assumptions of other (e.g., mixed model) approaches (see, e.g., Hubbard et al., 2010).
We agree that the somewhat arbitrary choice of two weeks for the effect lag is a limitation (and have better highlighted this fact in the discussion). We point out, though, that this can be viewed as part of the hypothesis - i.e., that there will be a differential impact of mode (RI vs NRI) in the community COVID incidence in two weeks – rather than an assumption per se. Thus, we have refrained from a sensitivity analysis (since, for example, we did not think that the impact of instructional mode would be seen in only one week). We agree with the implication that the true lag may be different than two weeks; a further investigation of this question would be of interest, though complex and beyond the scope of the paper. While the Reviewer’s suggested statistical approaches present interesting possibilities, we do believe that in our context that the use of the reproduction number, including the approach of Nash et al., would be rather complex (see, e.g., Delameter et al., 2019) and beyond the scope of the present paper.
Additional References:
Delamater, P. L., Street, E. J., Leslie, T. F., Yang, Y., & Jacobsen, K. H. (2019). Complexity of the Basic Reproduction Number (R0). Emerging Infectious Diseases, 25(1), 1-4.
https://doi.org/10.3201/eid2501.171901.
Hubbard AE, Ahern J, Fleischer NL, Van der Laan M, Lippman SA, Jewell N, Bruckner T, Satariano WA. To GEE or not to GEE: comparing population average and mixed models for estimating the associations between neighborhood risk factors and health. Epidemiology. 2010 Jul;21(4):467-74. doi: 10.1097/EDE.0b013e3181caeb90. PMID: 20220526.
Materials and Methods – Continuous exposure: Since school districts followed RI or NRI to a varying extent, have you also tested continuous exposures, e.g. the number of weeks with RI or NRI? Could be an additional/sensitivity analysis for the first analysis A1. Also, if you include all school districts, does a comparison between in-person, fully remove, and hybrid become feasible?
School districts implemented non-remote instruction in highly variable ways. Some hybrid models had children in school on alternate days, some a half day every day, some half weeks. Others had different schedules for different levels that varied for elementary vs high school. It varied over time even within a school district. It is very difficult to scale the mode exposure. However, at least some in-person time could be dichotomized and contrasted with no in-person time.
Results – SVI: Is the SVI dichotomization with a 0.5 cutoff common or are there empirically more justified cutoffs? Do you even need a cutoff or could you not just estimate the interaction between the continuous SVI and the binary instruction mode variable?
The SVI is a ranking and by using 0.5 as a cut off, we compare bottom half to upper half. We tested models with SVI as a continuous variable and it did not meaningfully change the estimates. Therefore, we chose to dichotomize the variable to make interpretation simpler.
Figures: Not sure what went wrong, but the quality of the figures is poor, especially for Figure 1 and 2, but also the others. I could only guess what was shown.
We have updated all of the figures and hope that helps with clarity.
Table 1: Could be moved to the appendix.
Table 1 has been moved to the appendix as recommended.
I hope my comments were helpful and I wish the authors best of luck with any revisions.
Thank you again for your thoughtful review. We hope these improvements and clarifications address your concerns.
Round 2
Reviewer 3 Report
Comments and Suggestions for Authors
I thank the authors for a revised version of their manuscript. Regrettably, I think further revisions are necessary as some of my original comments have not been fully addressed. I also disagree with some of the authors’ replies.
Materials and Methods – Analysis 1: I am sorry if my comment was not clear. I was suggesting to formally adjust for pre-study incidence. The results show lower incidence of RI zip codes during the study period but higher incidence of RI zip codes in the pre-study period compared to NRI zip codes. The way the results are presented and discussed seems to suggest that differences in incidence in the pre-study period could explain differences during the study period. However, the incidence in the pre-study period was generally low, so I am sceptical as to whether these differences can fully account for the differences during the study period when incidence was generally high. One way or another, it would be helpful to adjust for the pre-study incidence per zip code, to check whether any statistical association with RI/NRI during the study period remains. An adjusted analysis could render further comparison and discussion of the incidence in the pre-study period superfluous (e.g. line 262 to 269 of the revised manuscript).
Materials and Methods – Analysis 1: I am not convinced by the GEE approach and strongly advise against its justification (see edits in the revised version in line 146-150). By contrast, the GEE approach is a limitation, which should be mentioned at least in the Discussion. As it has been discussed here (https://link.springer.com/article/10.1007/s10654-022-00908-y) and here (https://www.thelancet.com/journals/lanpub/article/PIIS2468-2667(23)00046-4/fulltext), NPIs affect behavior which first and foremost impact transmission. Therefore, it is only reasonable to use a method that adequately adjusts for the stochastic delay between an unobserved outcome of transmission (such as the reproduction number) and the observed outcome (such as cases). GEE on cases with a fixed 2-week lag for the NPI is and always will be only a coarse alternative approach. I don’t think that the approach by Nash et al. to estimate the time-varying reproduction number from weekly case data is too complicated and “beyond the scope of the present paper”, as this approach is implemented in the extremely popular and easy-to-use EpiEstim package. If the authors stick to their GEE approach, they should at least acknowledge its limitations rather than justifying it as a “relevant” and “robust” approach, which I think it is not for this type of analysis.
Figure 2: I appreciate that the previous figures have become more readible, but it is lamentable that the new Figure 2 is again hardly readable. Also, I guess that in the figure no shaded area implies fully remove instructional mode? Please clarify in the caption. And why are there multiple grey lines but only a single blue area/line? I suggest to show all zip code incidences as lines.
Minor comments:
Line 202 and others: It can be confusing, that the numbers in the confidence intervals are separated just by a comma when the comma is used for numbers greater than 1,000 at the same time.
Figure 4: What is shown on the right axis? The number of zip codes?
Line 260: References should be provided for the following statement: “Surveillance data and studies have shown that high SVI has been associated with higher incidence and greater morbidity”.
Comments on the Quality of English Language-
Author Response
I thank the authors for a revised version of their manuscript. Regrettably, I think further revisions are necessary as some of my original comments have not been fully addressed. I also disagree with some of the authors’ replies.
Materials and Methods – Analysis 1: I am sorry if my comment was not clear. I was suggesting to formally adjust for pre-study incidence. The results show lower incidence of RI zip codes during the study period but higher incidence of RI zip codes in the pre-study period compared to NRI zip codes. The way the results are presented and discussed seems to suggest that differences in incidence in the pre-study period could explain differences during the study period. However, the incidence in the pre-study period was generally low, so I am sceptical as to whether these differences can fully account for the differences during the study period when incidence was generally high. One way or another, it would be helpful to adjust for the pre-study incidence per zip code, to check whether any statistical association with RI/NRI during the study period remains. An adjusted analysis could render further comparison and discussion of the incidence in the pre-study period superfluous (e.g. line 262 to 269 of the revised manuscript).
When developing the regression model utilizing GEE, we tested a model adjusting for the pre-study incidence, but it did not change the model, so we did not include it for simplicity. We have included the unadjusted RR for the study period along-side this regression model to provide a measure that is descriptive for the study period.
Lines 262-268 have been simplified – they are a descriptive summary of the data.
Materials and Methods – Analysis 1: I am not convinced by the GEE approach and strongly advise against its justification (see edits in the revised version in line 146-150). By contrast, the GEE approach is a limitation, which should be mentioned at least in the Discussion. As it has been discussed here (https://link.springer.com/article/10.1007/s10654-022-00908-y) and here (https://www.thelancet.com/journals/lanpub/article/PIIS2468-2667(23)00046-4/fulltext), NPIs affect behavior which first and foremost impact transmission. Therefore, it is only reasonable to use a method that adequately adjusts for the stochastic delay between an unobserved outcome of transmission (such as the reproduction number) and the observed outcome (such as cases). GEE on cases with a fixed 2-week lag for the NPI is and always will be only a coarse alternative approach. I don’t think that the approach by Nash et al. to estimate the time-varying reproduction number from weekly case data is too complicated and “beyond the scope of the present paper”, as this approach is implemented in the extremely popular and easy-to-use EpiEstim package. If the authors stick to their GEE approach, they should at least acknowledge its limitations rather than justifying it as a “relevant” and “robust” approach, which I think it is not for this type of analysis.
Thank you for the additional references and suggestions. We believe there is room for different philosophies regarding the approach, and as one of the references you provide mentions (Banholzer et al., abstract), “variation in methodologies may be desirable to assess the robustness of results”. Thus, it seems worthwhile for different approaches to be presented in the literature. Banholzer et al. mentions “non-mechanistic model”, including regression modeling, approaches, of which GEE would presumably be an example, and appears to suggest that there is a place for this class of approaches. GEE is a popular and well-regarded approach in many fields/applications; its potential advantages as well as disadvantages perhaps could use more attention in infectious disease epidemiology (and in particular, COVID-19 studies). We have already noted some of its limitations (the primary one being the fixed lag time). We would like to take the reviewer’s suggestion, while keeping the GEE approach, of elaborating further on its possible limitations (now in the Discussion), Furthermore, we have removed the words “appropriate” and “relevant” so as to avoid any suggestion on our part that this approach is the best one or without flaws. We are also now mentioning the possibility of using the reproduction number (e.g., Nash et al., 2023) method as an alternative approach (Discussion section, page 9): “A mechanistic modelling approach, for example focusing on the reproduction number, as in Nash et al., 2023, might be considered as an alternative, possibly more dynamic, way of addressing our study question.”
Figure 2: I appreciate that the previous figures have become more readible, but it is lamentable that the new Figure 2 is again hardly readable. Also, I guess that in the figure no shaded area implies fully remove instructional mode? Please clarify in the caption. And why are there multiple grey lines but only a single blue area/line? I suggest to show all zip code incidences as lines.
The lack of shading does correspond with fully remote instructional mode and that clarification has been added to the figure’s caption. Most school districts have only one included zip code. For the 5 school districts that include more than one zip code, area graphs have had borders and color variation added. The area graphs were maintained rather than all converted to line graphs to help give a better visual overview of which data contributed to the study. We have modified this figure in response to all reviewer comments. It attempts to visually synthesize trends over all zip codes.
Minor comments:
Line 202 and others: It can be confusing, that the numbers in the confidence intervals are separated just by a comma when the comma is used for numbers greater than 1,000 at the same time.
Confidence interval ranges have been modified to “-“
Figure 4: What is shown on the right axis? The number of zip codes?
Yes – (No.) has been added to the graph.
Line 260: References should be provided for the following statement: “Surveillance data and studies have shown that high SVI has been associated with higher incidence and greater morbidity”.
Thank you – the following reference has been added:
Dasgupta S, Bowen VB, Leidner A, et al. Association Between Social Vulnerability and a County’s Risk for Becoming a COVID-19 Hotspot — United States, June 1–July 25, 2020. MMWR Morb Mortal Wkly Rep 2020;69:1535–1541. DOI: http://dx.doi.org/10.15585/mmwr.mm6942a3